

# Preliminary results from the total lightning detector-cum-mini weather station installed at the Calcutta University

Subrata Kumar Midya[1], Sujay Pal[1], Reetambhara Dutta[1], Prabir Kumar Gole[1], Upal Saha[2], Goutami Chattopadhyay[1], and Soumyajit Hazra[3]

[1]Department of Atmospheric Sciences, University of Calcutta, Kolkata-700019
[2]National Centre for Medium Range Weather Forecasting, Noida-201309, India
[3]Skymet Weather Services Pvt. Ltd., Cyberone Tower, Sector-30 A, Vashi, Navi Mumbai

**Correspondence:** Sujay Pal (myselfsujay@gmail.com)

**Abstract.** We report preliminary results derived from the total lightning detector-cum-mini weather station installed at the Calcutta University during 2016. This detector is a part of Earth Networks Total Lightning Network (ENTLN) operated globally for ground-based monitoring of total lightning activity and forecasting of localized storm alert and severe weather conditions. This set up provides improved measurement of in-cloud (IC) lightning as well as cloud-to-ground (CG) lightning in addition to daily weather data. Severe weather such as thunder squall, Nor'wester, hailstorm, cyclone over the Gangetic West Bengal can be studied in details based on total lightning activity along with other atmospheric and meteorological research using the weather data. Here we present some initial results from the analysis of total lightning measurements during the recent Nor'wester events occurred in and around Kolkata. We also present variation of wet component of atmospheric refractivity index during the monsoon season which can be used to declare the onset and withdrawal time of monsoon over Gangetic West Bengal.

## 1 Introduction

Thunderstorm and lightning in the troposphere are the most significant atmospheric phenomena which keep life functioning on the earth and at the same time are incredibly destructive to human society in many ways. Lightning discharge radiates electromagnetic energy in a very wide radio frequency range, from below 1 Hz to near 300 MHz, with a maximum radiation energy in the frequency spectrum near 5 to 10 kHz (Rakov and Uman, 2003). Lightning discharge also radiates energy in the optical band $10^{14}$ to $10^{15}$ Hz visible to naked eye which enables ground based camera and satellite to take photograph of lightning event. Further, lightning produce X-rays and $gamma$-rays though not detectable at ground level. Because of this wide range of radiation frequency, there are many ground based and satellite based methods to monitor lightning activity in the atmosphere. Recently, there has been increasing interest in ground based lightning detection networks because of its potential use in meteorological applications. Lightning data from such lightning detection networks are useful for severe weather prediction such as high wind storms, tornadoes, flash flood, hailstorm etc. in addition to study of transient luminous events in the middle atmosphere. Continuous thunderstorm identification and tracking by lightning detection network also improves the nowcasts of thunderstorm, precipitation, severe weather, turbulence and tropical cyclone intensity which can act as a radar proxy in



areas of poor radar coverage (Liu and Heckman, 2011; Liu et al., 2014; Heckman et al., 2014). There are several ground based lightning detection networks operating globally for example, Earth Networks Total Lighting Network (ENTLN) which uses wideband electrical field recorders (1 Hz to 12 MHz), Worldwide Lightning Location Network (WWLLN) based on ground based VLF (3-30 kHz) detection of lightning sferics and European VLF/LF lightning detection network LINET.

In a cloud-to-ground discharge, electrical charge is effectively transferred from cloud to ground and normally known as CG lightning. CG lightning can be downward negative, upward negative, downward positive or upward positive discharge. Generally, about 90% or more global CG lightning are downward negative discharge and that of 10% or less of CG lightning are downward positive (Rakov and Uman, 2003). Electrical charge is not transferred from cloud-to-ground in all cases. In fact, majority of lightning discharge, almost 70% or more, occur within the cloud that do not involve ground, known as in-cloud (IC)

lightning which can be intra-cloud, inter-cloud and cloud-to-air discharge. In a thunderstorm, central and top parts of the cloud produce IC flashes during the initial stage of electrification, which can be enormously high for a severe storm (Williams et al., 1989). CG flashes increases with more storm electrification during active stage. Strong up-drafts during a storm produce high IC and positive CG flash rates which are the characteristics of a severe thunderstorm (Lang et al., 2000). Recent studies have shown that increase in total lightning i.e., IC and CG flash rate together can produce severe thunderstorm alert which generate

high wind, hail storm, tornadoes with a sufficient lead time ranging from 10 minutes to 1 hour (Liu and Heckman, 2011; Liu et al., 2014). In this paper, we have studied total lightning activity during two recent pre-monsoon summer thunderstorms, locally known as "Nor'wester", over the Gangetic West Bengal (GWB) around Kolkata which has not been done before. Nor'wester is the short duration severe thunderstorms with high wind speed occurring every year during late March to May in the eastern and north-eastern part of India including Bangladesh. This brings considerable damage to agriculture, properties

and even human life. Large number of lightning associated with the Nor'wester during the active stage of thunderstorm are the main reason of fatalities in this region. This paper attempts to study the total lightning activity during Nor'wester events from formation stage to dissipation stage with emphasis on short-term prediction of severity of the storm.

   We have also studied the variation of wet component of refractivity index during monsoon period using the data from the total lightning detector-cum-mini weather station (TLDWS) to find possible onset and withdrawal signature for monsson

over Kolkata. The monsoon specially south west monsoon is an important atmospheric circulation which affects the life and economy of Indian subcontinent. Traditionally monsoon is defined as the seasonal reversal of wind pattern associated with heavy precipitation. June, July, August and September are the principal monsoon months over Indian subcontinent. Indian summer monsoon rainfall (ISMR) is the rainfall carried by the south-west monsoon during June to September every year and accounts for approximately 80% of the annual rainfall over India. In recent years, people attach importance to study of

monsoon rainfall variations and proper prediction of onset and withdrawal of monsoon. Various studies represented the ISMR change is related to some meteorological parameters like surface temperature (Chattopadhyay et al., 1995), relative humidity, sea level barometric pressure (Parthasarathy et al., 1992; Bansod et al., 1995). El Nino Southern Oscillation (ENSO) events (Mooley et al., 1985; Gadgil et al., 2004), Sea Surface Temperature (Nicholls, 1995; Sahai et al., 2003; Rai and Pandey, 2008), Quasi Biennial Oscillation (QBO), Cloud condensation nuclei counter, aerosol concentration and even relative sunspot number

(SSN) and Flare index (Hiremath and Mandi, 2004) have significant impacts on monsoon. The accumulated impacts of these



various parameters make monsoon prediction more complicated and more challenging task. As monsoon is the principle rainy season over Indian sub-continent, proper prediction of onset and withdrawal of monsoon is very crucial. Climatologically monsoon onset takes place over Kerala (a Southern state in India) on 1st June and over Kolkata on 10th June. By the end of June, it covers more than 90% of the area of India and by mid-July the whole of India is covered by the monsoon. In early

September, summer monsoon rains begin to withdraw from north-west part of India and from entire country by mid-October. During monsoon onset, dramatic changes of large scale atmospheric structure are known to occur over India. Some of the well known ones associated with onset are rapid increase in daily rain rate, increase in the vertically integrated moisture and the increase in the strength of low level monsoon flow. Many researchers have studied onset and withdrawal of monsoon in India using various parameters, such as outgoing long wave radiation (OLR), integrated water vapour (IWP), low level jet

stream (LLJ), sea surface temperature (SST) (Joseph et al., 1994, 2006), wind data (Wang et al., 2009) and vertically integrated moisture transport variability (Fasullo and Webster, 2003). Using GPS radio occultation data for 2001-2010, Rao et al. (2013) examined variation of atmospheric refractivity during the onset of ISM over east Arabian sea and observed an enhancement of 5-10 N units in refractivity a few days before on set of monsoon over Kerala. Till today, scientists are trying to find out more and more reliable parameters for prediction of exact onset and withdrawal of monsoon. Midya et al. (2013a, b) reported that

the variation of wet component of refractivity gives an indication of cyclonic movement and onset of squall over Kolkata. In addition to study the total lightning activity during Nor'wester days, we have also examined the variation of wet component of refractivity, in this paper, during monsoon season to check possible signature of monsoon onset and withdrawal time.

## 2 Description of the detector

Earth Networks total lightning detector-cum-mini weather station (TLDWS), which was installed at the Department of At-

mospheric Sciences, University of Calcutta in July, 2016, consists of several parts (also shown in Figure 1) listed below: (a) Weather sensor which captures wind speed and direction; (b) Integrated in-cloud (IC) and cloud-to-ground (CG) lightning detection sensor which mainly measures electromagnetic signals from lightning discharge; (c) Sensor Shelter which measures mainly temperature, relative humidity, heat index, wind chill, barometric pressure and dew point; (d) Rain gauge which measures daily, monthly and yearly rainfall totals and averages; (e) Lightning remote box (f) Network appliance which is basically

an IP-enabled device that connects easily to the internet, provides fast transmission of the data to the server. It has a 72-hour battery life and automatically reboots as needed.

The integrated IC and CG lightning sensor operates in a frequency range from 1 Hz to 12 MHz (spanning the ELF, VLF, LF, MF, and HF ranges) and measures the electromagnetic signals from each lightning discharge (Heckman et al., 2014). The whole electric field waveforms are transferred from the censor to the central data processor of Earth Networks via internet and

network appliance. Central processor will then geolocate the individual lightning event and calculate the associated lightning parameters (such as peak current, multiplicity, differentiate lightning types etc.) from the waveform characteristics sends by this sensor and other sensors in the networks. Time of arrival method is being used to geolocate lightning event. In this method the onset time, arrival time, time of peak magnitude of a lightning pulse measured by multiple sensors (at least four) are analyzed



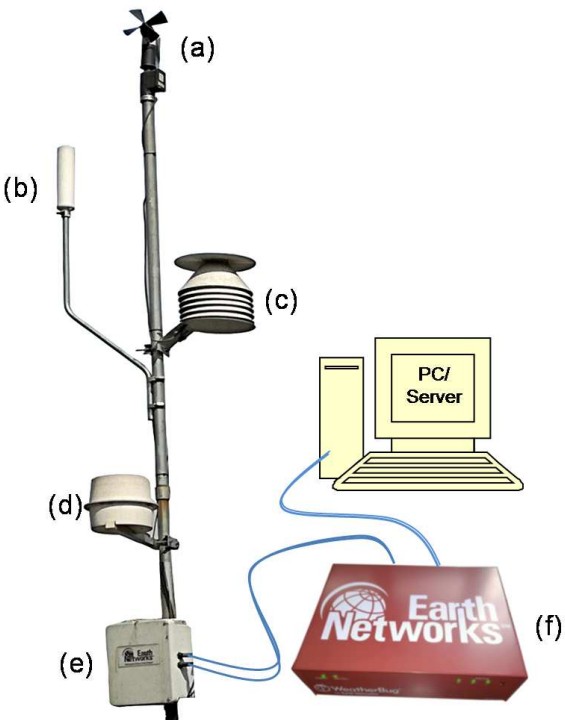

**Figure 1.** Earth Networks total lightning detector-cum-weather instrument: (a) Wind speed and direction sensor (b) Integrated in-cloud (IC) and cloud-to-ground (CG) lightning detection sensor (c) Sensor Shelter (d) Rain Gauge (e) Lightning Remote Box (f) Network appliance.

at the central processor to determine the four unknowns latitude, longitude, height and time that define the source location (Heckman, 2014). Each lightning discharge consists of several strokes. In the ENTLN, individual strokes occurring within 700 ms and 10 km of the first stroke detected by the sensor are clustered into a flash. A flash is further classified as a CG flash if it contains at least a return stroke, otherwise it is classified as a IC flash. In Figure 2, we have presented the total lightning count (IC+CG) per day for April, 2018 as detected by the TLDWS within its range covering the area between 87.65°E–89.52°E and 22.13°N–22.92°N). High lightning count greater than 10,000 are the Nor'wester days over this region.

## 3 Nor'wester and total lightning activity

Here we report the preliminary results of measurements of total lightning activity during pre-monsoon summer thunderstorms in and around Kolkata using the TLDWS. Total lightning activities corresponding to two Nor'wester events occurred on April 7 (event 1) and April 17, 2018 (event 2) are analysed. Event 1 was a non-severe type having maximum wind speed 64 km/hr with no causalities reported. Whereas around 15 people from Kolkata and adjacent districts were died (Source: The Hindu) during




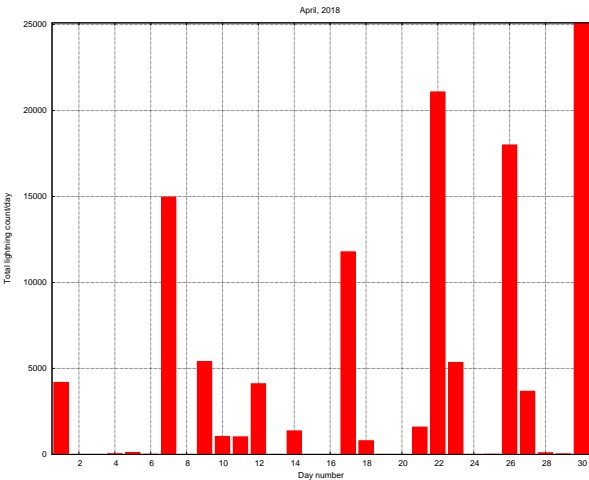

**Figure 2.** Total lightning flash count (IC+CG) per day as detected by the TLDWS for the month of April, 2018 within its range (87.65°E–89.52°E, 22.13°N–22.92°N).

the Nor'wester of 17th April with maximum wind speed of 98 km/hr recorded by IMD (Indian Meteorological Department), Kolkata between 19:42 IST to 20:15 IST.

The TLDWS at Calcutta University registered a total of 13,242 lightning flashes during 17:00 IST to 23:00 IST on April 7, of which 9,243 were in-cloud (IC) lightning (69.80 %) and 3,999 were cloud-to-ground (CG) lightning (30.20%). On April 17, TLDWS registered a total of 10,541 flashes during 17:30 IST to 22:00 IST, of which of which 7,403 were in-cloud (IC) lightning (70.24 %) and 3,138 were cloud-to-ground (CG) lightning (29.76%). Out of all CG flashes, almost 12% flashes were positive CG for the event 1, while 17% flashes were positive CG for the Nor'wester event 2 as classified by the TLDWS. Figure 3a and 3b show the evolution of different types of lightning flash rate (number of flashes per minute) as detected by the TLDWS within its range (around ~150 miles radius) for the event 1 and event 2 respectively. Black lines show CG flash rate, red line show IC flash rate and blue lines represent total lightning (CG+IC) flash rate per minute. Thick lines are the 7 point running averaged curves. For both events, total lightning flash rate increased drastically to about 110-120 flashes per minute during the active stage, from about 20-30 flashes per minute during the initial stage of thunderstorms showing lightning jump. Lightning jump in the flash rate are used to predict the severe thunderstorms well ahead of the peak damaging wind and hailstorm [Williams et al. 1999]. In Figure 3, dashed vertical lines are used to identify the time when Nor'wester hit the region with peak wind speed recorded by IMD, Kolkata. Therefore, it is obvious that study of total lightning activity can be a good indicator for the dangerous Nor'wester events well ahead of time to mitigate the damages caused by them.

The thunderstorm of 17th April ended with a sudden decrease in IC flash rate but the thunderstorm of 7th April shows gradual decay of both IC and CG flash rates. It is also to be noted that the CG flash rate peaked before the IC flash rate and damaging wind for the 2nd event, but the IC and CG flash rates simultaneously increased and decreased during the storm lifetime for the 1st event. Now to identify the severity of the thunderstorm using lightning characteristics, we have calculated



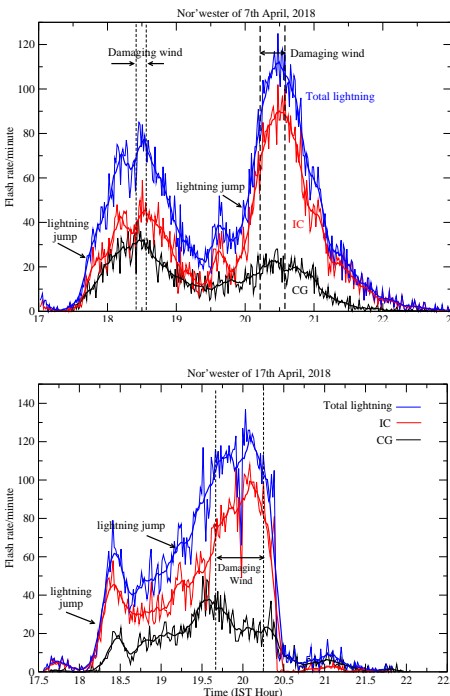

**Figure 3.** Temporal evolution of lightning flash rate per minute during two recent Nor'westers over Kolkata.

total discharge peak current per minute by the positive and negative CG flashes, since the CG flashes cause most damage to human life, power lines and consumer electronics on earth. Figure 4 shows the temporal distribution of total peak current per minute (kA/min) due to positive and negative CG flashes for the two events respectively. Solid lines are the 7 point running mean curves. While it is difficult to identify the intensity of the storm from the total peak current per minute from the negative

CG flashes, we can see that mean discharge current was more than 100 kA/min touching to 200 kA/min by the positive CG flashes during the active phase of thunderstorm on 17th April and the same for the thunderstorm of 7th April was merely 100 kA/min. Also for the 2nd event, 5% more positive CG flashes occurred than the 1st event. Therefore, positive CG lightning can be used to identify severity of thunderstorm event. We are analysing all the thunderstorm events over GWB based on total lightning activity and results will be reported in due time.

Another important characteristic of lightning is the stroke multiplicity which cause damage to human life and consumer electronics along with peak current. A lightning flash normally consist of one or several strokes which are within 700 ms and 10 km of the first stroke as detected by the sensor. The number of strokes in a flash is known as lightning multiplicity. Figure 5 shows the multiplicity distribution of negative and positive CG flashes occurred on April 7 and April 17, 2018 during the Nor'wester events. We note that 59% of negative CG flashes are composed of a single stroke for the thunderstorm of April

7, 2018 whereas 72% of negative CG flashes are composed of a single stroke for the severe thunderstorm of April 17, 2018.





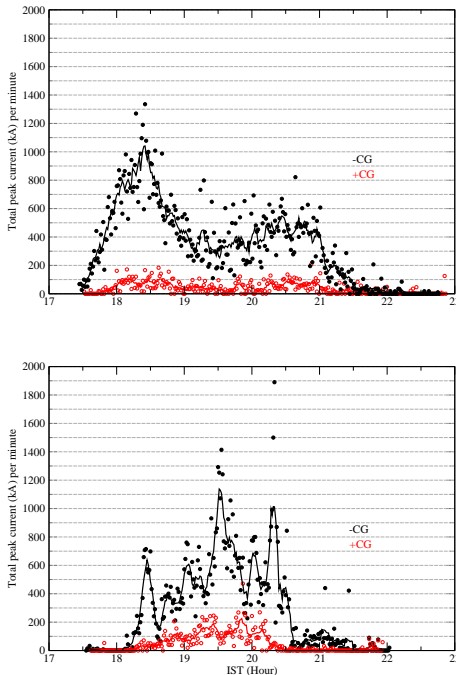

**Figure 4.** Temporal distribution of total peak current (kA) per minute due to ±CG lightning during the thunderstorms of 7th April and 17th April.

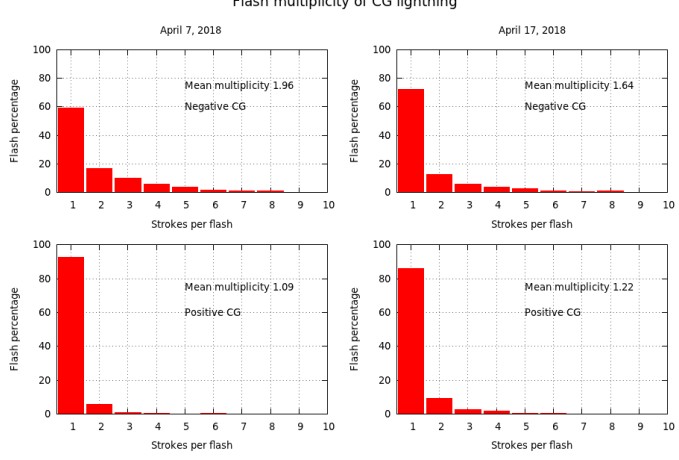

**Figure 5.** Flash multiplicity for positive and negative CG lightning flashes.



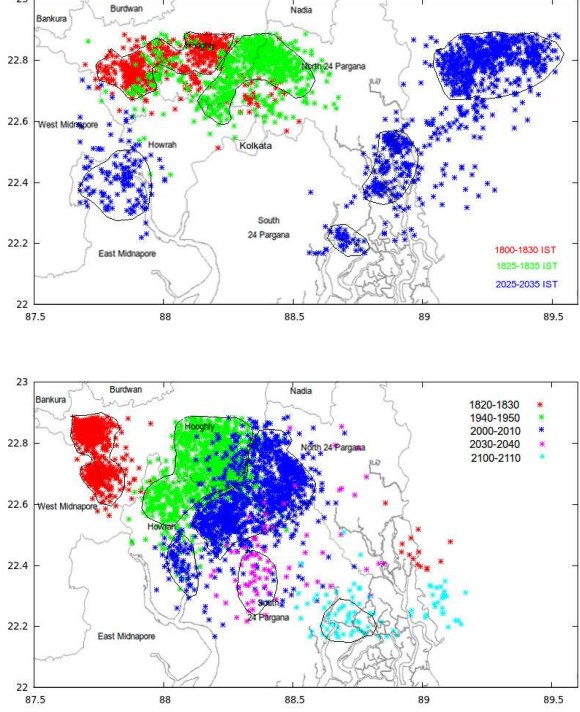

**Figure 6.** Temporal evolution of thunderstorm cell during the two Nor'wester events over Kolkata. Note that the thunderstorm cell comes from the north-westerly direction.

Average multiplicity of negative CG flashes are found to be 1.96 for April 7 and 1.64 for April 17, 2018, while the same for positive CG flashes are found to be 1.09 for April 7 and 1.22 for April 17, 2018 respectively.

It is also possible to track the life cycle of a thunderstorm cell using the total lightning activity which is frequently done by the weather radar system. When the lightning flash rates are high enough, thunderstorm cells can be identified with total flashes occurring in clusters which can be be used for early warning of severe storms [Betz et al. 2008; Liu et al. 2014]. In Figure 6, we present an example of time evolution of lightning cells as snapshots of total lightning activity for 10 minutes in the GWB region around Kolkata for both the thunderstorm events. Note that the thunderstorm cell comes from the north-westerly direction which gives its name as Nor'wester.

## 4   Atmospheric refractivity and monsoon over Kolkata

Here we present the variation of wet component of atmospheric refractivity index ($\eta_w$ in ppm) during monsoon period over Kolkata during 2016 using the data from TLDWS to show the potentiality of $\eta_w$ as one of the tool to declare the onset and withdrawal dates of monsoon. Variation of wet component of atmospheric refractivity with time has been studied using the


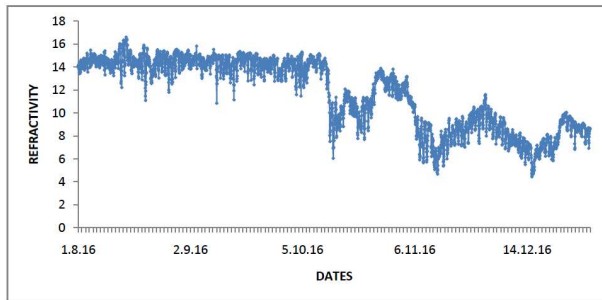

**Figure 7.** Variation of wet component of atmospheric refractivity index during August to December 2016 obtained from the TLDWS.

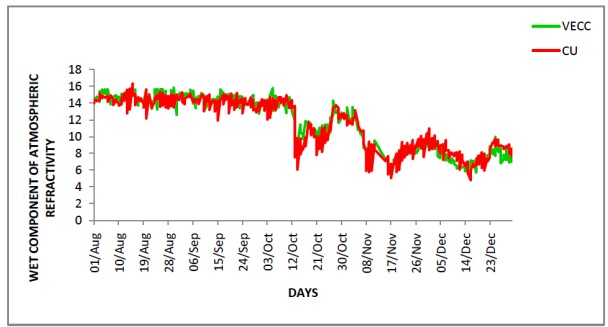

**Figure 8.** Comparison of wet component of atmospheric refractivity index obtained from the TLDWS and IMD (VECC), Kolkata. .

formula for troposphere as used by Midya et al. (2013a). Mainly atmospheric pressure, temperature and relative humidity are used to calculate ($\eta_w$). We have taken the wet component of refractivity because it is highly dependent on the presence of water vapour in the atmosphere. As the dry component of refractivity is not dependent on the presence of water vapour in the atmosphere, we have neglected this term. The variation of wet component of refractivity on hourly basis from August to

5 December 2016 is plotted in Figure 7. The figure shows a steady higher value of refractivity during monsoon and a sharp decrease in refractivity during withdrawal of monsoon over Kolkata. It can be also clearly seen that during the monsoon period the daily fluctuations in refractivity also reduced. Variation of wet component of refractivity from two data sources, the TLDWS and IMD, Kolkata (VECC station), are compared in Figure 8 and exactly the same variation is seen.

Monsoon is the reversal of wind pattern associated with heavy precipitation. In India two types of monsoon can be seen,

10 one is summer monsoon or south-west monsoon and another is winter monsoon or north-east monsoon. Gangetic West Bengal receives south-west monsoon dominantly, but north-east monsoon can hardly observe. In north hemispheric summer, due to differential heating of landmass and ocean body, a low pressure develops over interior of Asia as well as over North-Western India. At the same time a high pressure region persists over the Southern Indian Ocean. As a result winds blows from this high pressure region to low pressure region. After crossing the equator, due to Coriolis force, this wind turns into right and starts



flowing from the South-West direction and enters into Indian Peninsula. During the journey of this wind over warm Tropical Ocean, it acquires abundant moisture content within it. When it arrives near the southern tip of the Indian peninsula, this wind system breaks into two branches. One is the Arabian Sea branch which hits the Western Ghats, and another is the Bay of Bengal branch which flows over the Bay of Bengal and hits the eastern Himalaya. Thus during summer this surface westerly

wind (blowing from south west direction) brings ample amount of water vapor form the Bay of Bengal into the GWB basin. As Partial vapor pressure depends on relative humidity, wet component of refractivity noticeably increases and indicates the increased water vapour content in the atmosphere, when the moist air from Bay of Bengal enters in GWB Basin. When this water vapour condenses, heavy precipitation occurs in this region. As water vapour is the primary source of precipitation, the onset of monsoon is expected to occur over GWB when sufficient amount of water vapour has been carried from the Bay of

Bengal by the westerly wind into the GWB basin. Similarly when surface easterly blows dominantly, the amount of water vapour reduces over the GWB basin and monsoon is expected to withdraw. From Figure 7 it is seen that a sharp decrease in refractivity occurred on 13.10.16, so it may be monsoon withdrawal date in this year. IMD declares 16.10.2016 as the withdrawal date in GWB. IMD declares the dates on the basis of some criteria given later and in our study only wet component of refractivity is considered.

IMD which is the principal Government Department of Weather Forecasting declared onset and its further advancement over the country with three criteria given below. (a) Rainfall: If after 10th May, 60% of the available 14 stations viz. Minicoy, Amini, Thiruvananthapuram, Punalur, Kollam, Allapuzha, Kottayam, Kochi, Thrissur, Thalassery, Kannur, Kudulu and Mangalore report rainfall of 2.5 mm or more for two consecutive days, the onset over Kerala can be declared on the 2nd day provided the following criteria are also in concurrence. (b) Wind Field: Depth of westerlies should be maintained upto 600 hPa, in the

box equator to Latitude 10°N and Longitude 55°E to 80°E.The zonal wind speed over the area bounded by Latitude 5-10°N, Longitude 70-80°E should be of the order of 15-20 Kts. (c) Long wave radiation (OLR): INSAT derived OLR value should be below 200 $wm^{-2}$ in the box confined by Latitude 5-10°N and Longitude 70-75°E.

Withdrawal of monsoon are declared on the basis of reduction in moisture and prevalence of dry weather for 5 days. It is quite expected that the refractivity increases with increase of water vapour content in the atmosphere. During monsoon period,

when the water vapour enters over Gangetic West Bengal, refractivity increases significantly. Similarly when the water vapour is withdrawn over Gangetic West Bengal, monsoon disappears. But the fact is that, presence of water vapour is not only the criteria of onset of monsoon. Presence of cloud condensation nuclei (CCN), dew point temperature etc. are also essential criteria to start rainfall. In our observation refractivity becomes maximum when water vapour enters over GWB and during monsoon period it remains at a higher steady value because during whole monsoon period there is an ample amount of water

vapour supply. And during withdrawal of monsoon it sharply decreases from the higher steady value. Therefore, study of wet component of atmospheric refractivity can be used as a tool to declare the onset and withdrawal dates of monsoon.



## 5    Summary and Conclusion

Earth Networks total lightning detector-cum-mini weather station (TLDWS) has been installed at the Calcutta University aiming to monitor total lightning activity over the Gangetic West Bengal around Kolkata which could also be useful for early warning of severe weather along with weather and climate study in the region. We have discussed the performance

characteristics of the TLDWS and also compared the data with the IMD data. We have shown the usefulness of the system to predict the high and damaging wind corresponding to Nor'wester events around Kolkata. In our study, we have found that total lightning flash rate which includes both IC and CG flash rate starts increasing rapidly during the initial stage of the thunderstorm much before the high wind and high peak current CG lightning occurred. The severity of Nor'wester storm can also be predicted from the characteristics of IC lightning and positive CG lightning. More works are needed to establish the

relationship of total lightning characteristics with damaging wind, dangerous lightning, heavy rainfall and hailstorm associated with Nor'wester events in this region. We have also shown that onset and withdrawal date of Indian monsoon over Kolkata can be declared from the variation of atmospheric refractivity index obtained from the same set up. In summary, the TLDWS provides a very good opportunity to study severe weather associated with thunder squall, Nor'wester, hailstorm, cyclone, heavy precipitation as well as to study various other meteorological and atmospheric research in Gangetic West Bengal.

*Acknowledgements.* We acknowledge with thanks the Earth Networks to present the instrument to our University. Sujay Pal acknowledge the support from the University Grants Commission (UGC) under the Dr. D.S. Kothari Fellowship Scheme (No.F.4-2/2006(BSR)/ES/17-18/0007).



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
