# Peer review of "Preliminary results from the total lightning detector-cum-mini weather station installed at the Calcutta University"

_Natural Hazards and Earth System Sciences, 2018_

## Referee Comment (RC1) · Anonymous Referee #1 · 28 Aug 2018

The paper presents lightning activity in a region of interest using a lightning detector-cum-mini weather station. The analysis of lightning data includes both IC and CG lightning. The authors emphasized the importance of understanding of the detail characteristics of severe pre-monsoon thunderstorms over Gangetic West Bengal locally known as Nor'westers which bring considerable damages to human lives, properties, agriculture, power grids and electronics devices. The findings of the study could be used for early warning of such kind of thunderstorms. The study necessitates steps toward accurate warning of thunderstorms in the region for mitigation of damages caused by them and also focussed on the possibility of short-term prediction of the severity of the storm. More such studies within a network and long term monitoring could lead to

accurate warning.

The paper also presents variation of wet component of atmospheric refractivity to point out onset and removal of monsoon in the region. It is observed that refractivity remains at a higher level with less fluctuations during the period of monsoon but sharply decreases with significant fluctuations at the time of withdrawal of monsoon. This study of monsoon in the region is very relevant and crucial since with the monsoon starts the cultivating season.

Highlights:

» Initial results from the total lightning detector cum mini-weather station are presented.

» Total lightning characteristics of two pre-monsoon severe thunderstorms are studied here in details which has not been done before for the region.

» Total lightning count increased before peak wind and positive CG lightning count increased by 5% for the severe thunderstorm.

» Study of atmospheric refractivity index reveals monsoon arrival and withdrawal signature.

Comments:

–> Wind speed shown in Figure 3, whether they are geo-located or measured separately (how far from the thunderstorm cell?).

–> How is the peak wind speed related to lightning?

–> In the present analysis 17th April is chosen due to the severity of the thunderstorm on that day. According to Figure 2, there were more lightning activity on 22nd, 26th and 30th April. How was the wind character on those days?

–> Sharp decline in the refractivity plot is considered in this paper as the withdrawal signature of the monsoon. But the variations of refractivity plot does not indicate onset

of monsoon. It is suggested to plot the variations over the whole year. Also, if possible, can include a few year's data for the definitive conclusion of the withdrawal signature.

–> Increase the text size of the labels and titles in the figures from 2 to 6.

–> In Figure 3, Top and down panels are slightly overlapped.

---

## Referee Comment (RC2) · Anonymous Referee #2 · 10 Sep 2018

The manuscript describes lightning detection data from an Earth Networks TLDWS station during two "Nor'wester"-type thunderstorms affecting the Kolkata region in April 2018. An additional component of the manuscript describes Monsoon onset and departure monitoring using the wet component of atmospheric refractivity index.

Lightning section comments: Data produced by Earth Networks from the two storm case studies are presented and discussed by the authors. Presentation and summary are adequate. Whilst new data are presented, they do not provide insight which has not already been provided by other researchers on the subject of lighting flash rate and characteristics during the active phase of severe thunderstorms. Whilst it is accepted

that the paper addresses preliminary results and more data are needed for firm conclusions, the confidence of any causal relationship between lightning and the onset of damaging wind based on two storms must be rather low and therefore should still be considered as inconclusive, albeit consistent with current research on this topic. Despite this, it is valuable to have observational data on "Nor'wester" severe storms which are relatively underrepresented in thunderstorm research literature, so the continued collection of information on these would become valuable once a longer dataset capable of producing more statistically robust characteristics on these storms could be assessed.

Refractivity section comments: Although previous researchers have used refractivity wet component to investigate thunderstorm activity in the region, it is unclear why the authors decided to use this complex derived parameter, given their instrumentation measured the fundamental atmospheric properties of temperature, pressure and humidity and microwave propagation was not declared to be of interest for this work. If the variability of near-surface moisture content was considered to be useful for the monitoring of the monsoon, why not simply use dew-point temperature? Both dew-point and refractivity are a function of vapour pressure and variability of air pressure could be examined independently if it was considered to be of merit. The finding that refractivity (e.g. humidity) reduces after the monsoon is a somewhat expected outcome which could be adequately explained using more fundamental considerations.

Spelling/grammar and minor comments: p.1 line 8: events that occurred p.3 line 29: replace censor with sensor p.3 line 31: replace sends with sent p.3 line 32: replace networks with network p.4 line 11: replace died with killed p.6 line 10: Please provide a reference to support multiplicity as an important factor in lightning damage. p.6 line 11: replace consist with consists p.8 line 11: replace potentially with potential p.8 line 11: replace tool with tools p.10 line 27: Surely CCN availability cannot be considered a limiting factor for monsoon rainfall generation? evidence to support this idea would be needed to maintain this suggestion.

To summarise, whilst the manuscript does not currently provide sufficiently robust or original insight, the objective of gathering more data on Nor'wester-type thunderstorms is valuable and its continued collection, analysis and eventual publication should be greatly encouraged.

---

## Referee Comment (RC3) · Anonymous Referee #3 · 26 Oct 2018

The paper is basically interesting, when one assumes that the area has never been examined in terms of lightning activity (please provide references on local lightning in the area, if existing). Apparently, the authors have received a set of lightning data and analysed the stroke occurrence in various ways. However, the impression is given that all that emerges from a single sensor. In fact, a large network has been used, but this system is not described and the functional working of the sensors remains in the dark. Most important, signals from a huge frequency range are claimed to enter into the analysis, performed by an unknown party, while it remains completely unresolved how the basically different signals from the various physical sources are treated and put together, or whatever else has been done.

[Figure]

The data handling procedures must be described, otherwise the reader cannot assess the significance, or compare with other systems. Substantial rewriting is required.

Some comments are given in detail:

1. Abstract, p. 1, line 1: notes a "preliminary" report. Does it mean that the data is not yet fully correct or presently too limited to attribute significance to the results?

2. Abstract, p.1, line 1: notes that the authors present lightning data results from a single sensor placed in Calcutta. This is highly misleading, because it is not a "stand-alone" that could deliver lightning data. It is also noted that this sensor is part of a network; thus, the lightning data comes from this network, not from a Calcutta sensor. This should be clearly clarified.

3. Introduction: it should be mentioned what kind of lightning data examination has been performed for the relevant area in the past.

4. P. 3, Line 27: the measuring system is not described adequately. For example, it is not said, what kind of discharge events are identified. For any other network the manufacturer or user describe that CG strokes or IC strokes (centered around 10 kHz), or source points (or leader steps) in the VHF range are measured. Leader steps are always present when a channel forms in flashes that may remain in the clouds or contact ground. Thus, a VHF 'signal' cannot be attributed to either a cloud or a CG flash. The noted ref. "Heckmann et al. 2014" does not present any explanation along these lines. As a result, the reader does not know what is really measured and how it is interpreted.

5. P. 2, lines 2-3, P. 3, Line 27: the authors claim that the system uses signals down to 1 Hz. This is not credible and sounds quite absurd. Even Schuman resonances start at higher frequencies, and it is totally clear the simple rod (fig. 1) is not suited to detect Schuman resonances; even though, handling and evaluation of ELF requires quite different methods than those that seem to be used by the present sensor data

analysis.

6. P. 3, line 31: the authors state that the 'signals', presumably including the VHF signals, are processed in order to give current, multiplicity and lightning type. This needs more explanations. First, the quoted parameters are not relevant for VHF signals; a source point (leader step signal) may have any strength and is basically not associated with the peak current of a return stroke or an IC-stroke in the VLF/LF range. A VLF/LF stroke may be CG or IC; the procedure to distinguish needs to be explained, because different methods are in use in other networks and it is known to be quite difficult and often ambiguous.

7. P. 4, line 1: the sensor signals are used to locate 'sources'. What is meant by 'sources'? Traditionally, sources are VHF events; do the authors mean VHF or VLF/LF events?

8. P. 4, line 2: it is correct that discharges may produce strokes, either CG or IC strokes. However, these strokes are exclusively VLF/LF events and can not be determined by VHF signals. Thus, VHF signals should be excluded in this consideration. When strokes are grouped into a flash, as described, only CG strokes can be taken, because IC discharges extend quite often over more than 10 km horizontal distance and last longer than 700 ms. But when VHF is excluded here, where are these VHF data handled and shown? This treatment of the measured signals remains totally unexplained. Thus, this part of the "detector description" (as the section is headed) needs substantial rewriting.

9. P. 3, Chapter 2: the "description of the detector" is insufficient. The chapter must include an understandable description of the handling of the data from the various frequency ranges. Naturally, the network configuration must also be described, the number of used sensors and the relevant baselines should be given.

10. It remains unclear from where the lightning data comes. The signals from the Calcutta station are insufficient. It should be explained that – as I assume - the network

owner provides processed data to the authors, i.e. stroke or event listing. Insofar, also the acknowledgement is misleading. 11. Fig. 2: it is claimed that strokes, grouped to flashes, are shown. Again, the question arises how VHF signals are taken into account. The reader can not understand what the authors have really plotted.

12. P. 4, line 7: in extension of the previous points, the term "total lightning" needs an explanation. Presumably, it is not just the combination of CG and IC strokes in the VLF/LF range, because VHF signal somehow contribute in a totally unexplained manner. Finally, the question arises, how 1 Hz signals, or ELF signals contribute. In the opinion of this referee, ELF does not matter at all, but it is the authors obligation to communicate in full the used techniques and procedures, and to remove misleading or unnecessary parts.

13. P. 6, line 1: it does not make much sense to add all peak currents of all CG strokes in a storm, because the strokes occur independent of each other at very different locations and the size of the cells may largely vary. It suffices to characterize storm severity by the number of strokes per time and per area.

14. P. 6, line 13 and Fig. 5: the multiplicity needs a word on the lower threshold of currents that are determined. The authors should show an additional graph with the distribution of currents for the storm.

15. P. 11, line 1, chapter 5, Conclusion: again, it is not acceptable to claim that installation of one single station allows for monitoring total lightning. In fact, a large number of stations is required and the shown results could also be obtained without the Calcutta station. In total, the paper needs substantial rewriting, because the used instruments, analysis specifications and data handling are not described, preventing the reader from understanding what has been done.

---

## Author Comment (AC1) · 7 Dec 2018

**Interactive comments on "Preliminary results from the total lightning detector-cum-mini weather station installed at the Calcutta University" by Midya et al.**

We thank the reviewers for their minute scrutiny of the manuscript. We believe that their valuable comments and suggestions will be helpful for further improvements of the quality and credibility of the paper. We have revised the manuscript considering their suggestions. Answers are given below in red. Changes in the revised version of the paper are in bold black font.

**Reviewer 1:**

**The paper presents lightning activity in a region of interest using a lightning detector-cum-mini weather station. The analysis of lightning data includes both IC and CG lightning. The authors emphasized the importance of understanding of the detail characteristics of severe pre-monsoon thunderstorms over Gangetic West Bengal locally known as Nor'westers which bring considerable damages to human lives, properties, agriculture, power grids and electronics devices. The findings of the study could be used for early warning of such kind of thunderstorms. The study necessitates steps toward accurate warning of thunderstorms in the region for mitigation of damages caused by them and also focussed on the possibility of short-term prediction of the severity of the storm. More such studies within a network and long term monitoring could lead to accurate warning.**

**The paper also presents variation of wet component of atmospheric refractivity to point out onset and removal of monsoon in the region. It is observed that refractivity remains at a higher level with less fluctuations during the period of monsoon but sharply decreases with significant fluctuations at the time of withdrawal of monsoon. This study of monsoon in the region is very relevant and crucial since with the monsoon starts the cultivating season.**

**Highlights:**

**» Initial results from the total lightning detector cum mini-weather station are presented.**
**» Total lightning characteristics of two pre-monsoon severe thunderstorms are studied here in details which has not been done before for the region.**
**» Total lightning count increased before peak wind and positive CG lightning count increased by 5% for the severe thunderstorm.**
**» Study of atmospheric refractivity index reveals monsoon arrival and withdrawal signature.**

**Comments:**
**Wind speed shown in Figure 3, whether they are geo-located or measured separately (how far from the thunderstorm cell?).**

Ans: Wind speed data are collected from the Regional Meteorlogy Centre, IMD, Alipore, Kolkata. We have not measured the distance of the IMD station from the thunderstorm cell which produced the peak wind at corresponding times. But from the Figure 4&7, it can be noted the relative position of the lightning cells and IMD station. Important point to note here is the occurrence time of the peak/damaging wind speed after the lightning jumps.

**How is the peak wind speed related to lightning?**

Ans: The relation between peak wind speed and lightning are not established here as it requires more data (which can be done in future). What we focused here is the occurrence of lightning jumps before damaging winds which can be used as precursors of severe weather.

**In the present analysis 17th April is chosen due to the severity of the thunderstorm on that day. According to Figure 2, there were more lightning activity on 22nd, 26th and 30th April. How was the wind character on those days?**

Ans: We have analysed and compared the two events here. No wind gust was recorded for thunderstorm on 22nd April, however peak winds produced associated with the thunderstorms on 26th and 30th April were 80 km/hr and 60 km/hr respectively as recorded in IMD stations at Kolkata. But we did not mention these in the manuscript as we have also not analysed those events here. A statistical analysis of all the thunderstorms events of 2018 are being done and will be communicated shortly.

**Sharp decline in the refractivity plot is considered in this paper as the withdrawal signature of the monsoon. But the variations of refractivity plot does not indicate onset of monsoon. It is suggested to plot the variations over the whole year. Also, if possible, can include a few year's data for the definitive conclusion of the withdrawal signature.**

Ans: TLDWS was operating from July, 2016, so we do not have the data before that. However, we have extended the graph (Figure 9) for the whole year to show the refractivity variation from the IMD station where the monsoon onset part can be seen after June (day number 180). We have also revised the Figure 8 to include the variation of surface temperature, relative humidity and water vapor pressure during July, 2016 to December 2016.

**Increase the text size of the labels and titles in the figures from 2 to 6.**

Ans: Revised figures with increased (20%) label and titles are included now.

**In Figure 3, Top and down panels are slightly overlapped.**

Ans: We fixed it in the revised version.

Considering the comments and suggestions of the third reviewer, we have slightly revised the title of the manuscript as following:
"A preliminary study on thunderstorms and monsoon using total lightning and weather data over Gangetic West Bengal".

---

## Author Comment (AC2) · 7 Dec 2018

We thank the reviewers for their minute scrutiny of the manuscript. We believe that their valuable comments and suggestions will be helpful for further improvements of the quality and credibility of the paper. We have revised the manuscript considering their suggestions. Answers are given below in red. Changes in the revised version of the paper are in bold black font.

**Reviewer #2:**

**The manuscript describes lightning detection data from an Earth Networks TLDWS station during two "Nor'wester"-type thunderstorms affecting the Kolkata region in April 2018. An additional component of the manuscript describes Monsoon onset and departure monitoring using the wet component of atmospheric refractivity index.**

**Lightning section comments: Data produced by Earth Networks from the two storm case studies are presented and discussed by the authors. Presentation and summary are adequate. Whilst new data are presented, they do not provide insight which has not already been provided by other researchers on the subject of lighting flash rate and characteristics during the active phase of severe thunderstorms. Whilst it is accepted that the paper addresses preliminary results and more data are needed for firm conclusions, the confidence of any causal relationship between lightning and the onset of damaging wind based on two storms must be rather low and therefore should still be considered as inconclusive, albeit consistent with current research on this topic. Despite this, it is valuable to have observational data on "Nor'wester" severe storms which are relatively underrepresented in thunderstorm research literature, so the continued collection of information on these would become valuable once a longer dataset capable of producing more statistically robust characteristics on these storms could be assessed.**

**Refractivity section comments: Although previous researchers have used refractivity wet component to investigate thunderstorm activity in the region, it is unclear why the authors decided to use this complex derived parameter, given their instrumentation measured the fundamental atmospheric properties of temperature, pressure and humidity and microwave propagation was not declared to be of interest for this work. If the variability of near-surface moisture content was considered to be useful for the monitoring of the monsoon, why not simply use dew-point temperature? Both dew-point and refractivity are a function of vapour pressure and variability of air pressure could be examined independently if it was considered to be of merit. The finding that refractivity (e.g. humidity) reduces after the monsoon is a somewhat expected outcome which could be adequately explained using more fundamental considerations.**

**To summarise, whilst the manuscript does not currently provide sufficiently robust or original insight, the objective of gathering more data on Nor'wester-type thunderstorms is valuable and its continued collection, analysis and eventual publication should be greatly encouraged.**

Ans: We agree with the reviewer that collection of total lightning data corresponding to more Nor'wester type thunderstorms and analysis are very necessary to understand the lightning-thunderstorm dynamics, as the frequency of violent thunderstorms is increasing now-a-days causing a lot of damages to properties, agriculture and economy.

In the revised manuscript, we have added, as an example, the electric field waveform data from Kolkata station to show the raw measurements of the lightning sensor and identified the four types of discharge (+IC, -IC, +CG, -CG) which somehow helps to understand the measurements of the lightning sensor. We have also added the lightning location maps corresponding to the events of 7[th] April and 17thApril over the study region which includes all the lightning data that were used to

plot the flash rate graphs. As per suggestion of one of the reviewer, we have included previous research reports, as available in the literature, on lightning and thunderstorms over the area under investigation. We have substantially revised the manuscript and also changed the title as following: "A preliminary study on thunderstorms and monsoon using total lightning and weather data over Gangetic West Bengal".

About refractivity section:
We agree with the reviewer in this point and therefore added the more fundamental parameters like surface temperature, relative humidity and water vapor pressure in our analysis. Both vapor pressure and wet component of refractivity index are functions of temperature and humidity. It is seen that variability in the vapor pressure during the monsoon is comparatively lower than that of relativity humidity and both the parameters can be used to study further about monsoon onset and withdrawal processes.

**Spelling/grammar and minor comments:**
**p.1 line 8: events that occurred >> corrected**
**p.3 line 29: replace censor with sensor >> corrected**
**p.3 line 31: replace sends with sent >> corrected**
**p.3 line 32: replace networks with network >> corrected**
**p.4 line 11: replace died with killed >> corrected**
**p.6 line 10: Please provide a reference to support multiplicity as an important factor in lightning damage. >> We have added the following references in the revised manuscript.**

> **Gibbs, Alexander R., "Periodicities of Peak Current and Flash Multiplicity in Cloud to Ground Lightning" (2012). Dissertations & Theses in Earth and Atmospheric Sciences. 24.** http://digitalcommons.unl.edu/geoscidiss/24

> **Miyazaki, T., S. Okabe, 2008: A Detailed Field Study of Lightning Stroke Effects on Distrubtion. IEEE Transactions on Power Delivery, 24, 352-359.**

**p.6 line 11: replace consist with consists >> corrected**
**p.8 line 11: replace potentially with potential >> corrected**
**p.8 line 11: replace tool with tools >> corrected**
**p.10 line 27: Surely CCN availability cannot be considered a limiting factor for monsoon rainfall generation? evidence to support this idea would be needed to maintain this suggestion. >> We claim that the presence of water vapour is not only a criterion of onset of monsoon, but presence of cloud condensation nuclei (CCN) and dew point temperature play a significant role to start rainfall. The following references are included in the revised manuscript.**

> Midya S. K., Ghosh S., Ganda S.C., and Das, G.K.: Role of Biogenic Hydrocarbon on the Variability of Total Rainfall Amount over Sundarban, Kaziranga and Gir Forests, J. Ind. Geo. Union, 19, 454--459, 2015.

> Ganda, S.C., and Midya, S.K.: Comparison of long term rainfall trends on urban and non-urban regions of Indian land mass and its probable impliction, J. Ind. Geo. Union, 16(2), 37--40, 2012.

---

## Author Comment (AC3) · 7 Dec 2018

We thank the reviewers for their minute scrutiny of the manuscript. We believe that their valuable comments and suggestions will be helpful for further improvements of the quality and credibility of the paper. We have revised the manuscript considering their suggestions. Answers are given below in red. Changes in the revised version of the paper are in bold black font.

Reviewer 3:

The paper is basically interesting, when one assumes that the area has never been examined in terms of lightning activity (please provide references on local lightning in the area, if existing). Apparently, the authors have received a set of lightning data and analysed the stroke occurrence in various ways. However, the impression is given that all that emerges from a single sensor. In fact, a large network has been used, but this system is not described and the functional working of the sensors remains in the dark. Most important, signals from a huge frequency range are claimed to enter into the analysis, performed by an unknown party, while it remains completely unresolved how the basically different signals from the various physical sources are treated and put together, or whatever else has been done. The data handling procedures must be described, otherwise the reader cannot assess the significance, or compare with other systems. Substantial rewriting is required.

Ans: We clearly mentioned that the thunderstorm events in the area under investigation have never been examined with respect to **total lightning** (i.e., considering CG and IC at a time), at least we did not find any material in existing literature. We have added references from previous research reports on lightning activity in the area mentioning the data used in the analysis. We have revised the manuscript thoroughly to remove the misconception that the total lightning data emerges from a single sensor. We have also changed the title of the manuscript to avoid this misunderstanding. The Earth Networks total lightning network (ENTLN) is actually working with many more receivers in the area and lightning location and other parameters are determined at the central processor from the waveforms data sent by these sensors. The ENTLN is unique compared to other existing sensor technologies as they claim (Heckman and Liu, 2010 and other reports from Earth Networks). The sensor is a wideband system with detection frequency ranging from 1HZ to 12MHZ. ENTLN claims that this wide frequency range enables the sensor to not only detect strong CG strokes, but to also detect weak IC pulses beyond the line of sight. The primary focus of the ENTLN appears to maximize the detection efficiency for cloud flashes.

We have included this description in the revised manuscript and also presented the raw measurements from the sensor at Kolkata. These waveforms in raw digitizer units are converted to electric field unit (V/m) based on sensor calibration data and used to determine the location using the well known time of arrival (TOA) method. Waveforms data from minimum four sensors give the lightning location and other parameters. We have analysed the total lightning data set obtained from the ENTLN for the area under investigation bounded by the geographic area $87.65^0$ E–$89.52^0$E and $22.13^0$ N–$22.92^0$ N.

We hope the revised version will have no misconception about the data and methodology.

Some comments are given in detail:

1. Abstract, p. 1, line 1: notes a "preliminary" report. Does it mean that the data is not yet fully correct or presently too limited to attribute significance to the results?

Ans: Here the "preliminary" report means only initial report not necessarily contains detail analysis using large data set. The word "preliminary" has nothing to do with data quality or the results in the paper. In literature there are many research papers which include the phrase "preliminary results/ preliminary report/ preliminary study" etc. even in title.

2. Abstract, p.1, line 1: notes that the authors present lightning data results from a single sensor

placed in Calcutta. This is highly misleading, because it is not a "standalone" that could deliver lightning data. It is also noted that this sensor is part of a network; thus, the lightning data comes from this network, not from a Calcutta sensor. This should be clearly clarified.

Ans: We have revised the abstract (and also the title) to avoid any misconception or misunderstanding about total lightning data. Truly the lightning sensor is not standalone, only the weather station is standalone. We clearly mention this in revised manuscript and also in the abstract.

3. Introduction: it should be mentioned what kind of lightning data examination has been performed for the relevant area in the past.

Ans: We have added references from previous research reports on lightning activity in the area as available in the literature. We also mention the data used in their analysis.

4. P. 3, Line 27: the measuring system is not described adequately. For example, it is not said, what kind of discharge events are identified. For any other network the manufacturer or user describe that CG strokes or IC strokes (centered around 10 kHz), or source points (or leader steps) in the VHF range are measured. Leader steps are always present when a channel forms in flashes that may remain in the clouds or contact ground. Thus, a VHF 'signal' cannot be attributed to either a cloud or a CG flash. The noted ref. "Heckmann et al. 2014" does not present any explanation along these lines. As a result, the reader does not know what is really measured and how it is interpreted.

Ans: The ENTLN is unique compared to other existing sensor technologies as they claim (Heckman and Liu, 2010 and other reports from Earth Networks). The sensor is a wideband system with detection frequency ranging from 1HZ to 12MHZ. Primary focus of the ENTLN appears to maximize the detection efficiency for cloud flashes. The ENTLN claims to detect weaker pulses at longer distances than other VLF/LF systems with similar baselines by extending the frequency range of detection into the MF and HF spectrums. Thus ENTLN measures both IC and CG strokes. The strokes are grouped into a flash if they occur within 700 ms and 10 km of the first stroke detected by the sensors. A flash is further classified as a CG flash if it contains at least a return stroke, otherwise it is classified as IC flash. This is mentioned in the manuscript and we have analysed the lightning flashes which are classified as either +/-IC or +/- CG in the manuscript. We have not analysed the individual strokes which can be done with electric field waveforms data from the sensors.
We have added extra description of the total lightning sensor and also added raw electric field waveform data as recorded at Kolkata station identifying four types of lightning discharge events with distances from the sensor.
"Heckman, 2014: ENTLN Status Update, XV International Conference on Atmospheric Electricity, 15-20 June 2014, Norman, Oklahoma, U.S.A." clarifies in some extent the lightning location technique.

[Figure]

Top panel shows the raw measurements of electric field amplitude data for 1 minute time interval after 14:15 UT of 17[th] April, 2018. Electric field amplitude here is in raw digitizer units. Other four panels show the variation of electric field waveform corresponding to four types of lightning discharge as identified by ENTLN. Distances from the lightning location to the receiving station corresponding to the four discharges are 50.50 km (for +CG), 43.80 km (for -CG), 43.89 km (for +IC) and 7.80 km (for -IC) respectively. Note the presence of stepped leader (SL) and return stroke (RS) in case of CG lightning discharge.

5. P. 2, lines 2-3, P. 3, Line 27: the authors claim that the system uses signals down to 1 Hz. This is not credible and sounds quite absurd. Even Schuman resonances start at higher frequencies, and it is totally clear the simple rod (fig. 1) is not suited to detect Schuman resonances; even though, handling and evaluation of ELF requires quite different methods than those that seem to be used by the present sensor data analysis.

Ans: Well in that case we wish to refer the Earth Network lightning location principle, their manuals and research papers that used ENTLN data.
By extending the frequency ranges in both low and high frequency domain, ENTLN claims to detect and report weaker pulses at longer distances and achieve greater efficiency. But this is yet under-presented in the literature.

6. P. 3, line 31: the authors state that the 'signals', presumably including the VHF signals, are processed in order to give current, multiplicity and lightning type. This needs more explanations. First, the quoted parameters are not relevant for VHF signals; a source point (leader step signal) may have any strength and is basically not associated with the peak current of a return stroke or an IC-stroke in the VLF/LF range. A VLF/LF stroke may be CG or IC; the procedure to distinguish needs to be explained, because different methods are in use in other networks and it is known to be quite difficult and often ambiguous.

Ans: This is also mentioned in the previous comment that the ENTLN is unique compared to other

existing sensor technologies as they claim (Heckman and Liu, 2010 and other reports from Earth Networks). The sensor is a wideband system with detection frequency ranging from 1HZ to 12MHZ. primary focus of the ENTLN appears to maximize the detection efficiency for cloud flashes. The ENTLN claims to detect weaker pulses at longer distances than other VLF/LF systems with similar baselines by extending the frequency range of detection into the MF and HF spectrums. Thus ENTLN measures both IC and CG strokes. The strokes are grouped into a flash if they occur within 700 ms and 10 km of the first stroke detected by the sensors. A flash is further classified as a CG flash if it contains at least a return stroke, otherwise it is classified as IC flash. This is mentioned in the manuscript and we have analysed the lightning flashes which are classified as either +/-IC or +/- CG in the manuscript.

7. P. 4, line 1: the sensor signals are used to locate 'sources'. What is meant by 'sources'? Traditionally, sources are VHF events; do the authors mean VHF or VLF/LF events?

Ans: Here the source does not mean exactly the whole lightning channel, because the purpose of the network is not to image the lightning channel itself but to overall determine the location usually interpreted as some approximation to the ground strike point. The best electromagnetic channel imaging methods at VHF and the best ground strike-point locating techniques at VLF and LF have accuracies (actually location errors or uncertainties) of the order of a hundred meters. ENTLN uses a wide frequency range for their purpose.

8. P. 4, line 2: it is correct that discharges may produce strokes, either CG or IC strokes. However, these strokes are exclusively VLF/LF events and can not be determined by VHF signals. Thus, VHF signals should be excluded in this consideration. When strokes are grouped into a flash, as described, only CG strokes can be taken, because IC discharges extend quite often over more than 10 km horizontal distance and last longer than 700 ms. But when VHF is excluded here, where are these VHF data handled and shown? This treatment of the measured signals remains totally un-explained. Thus, this part of the "detector description" (as the section is headed) needs substantial rewriting.

Ans:  This is not correct. The signals produced by most cloud flashes become comparable to those produced by ground flashes only at higher frequencies. The ENTLN just extends the operating range to the high frequency bands to add improved detection of cloud flashes to the system's ability to detect ground flashes. Generally areas with low sensor density favor CG lightning detection (Liu and Heckman, 2010; Thompson et al., 2014). And in our analysis we analysed the IC and CG flashes as determined by the ENTLN using the whole frequency bands.

9. P. 3, Chapter 2: the "description of the detector" is insufficient. The chapter must include an understandable description of the handling of the data from the various frequency ranges. Naturally, the network configuration must also be described, the number of used sensors and the relevant baselines should be given.
 Ans: The actual number of sensors in the Earth Networks, operating in the area under investigation is not known to us. The network uses many sensors for their purpose. In the Gangetic West Bengal region, locations of four sensors including Kolkata station are known to us. There could be more sensors. Actually the lightning location is determined using many sensors, but we do not know the locations of all sensors. We are focusing only to use ENTLN data for our research purpose. Therefore, the description of network configuration in this region is beyond the scope of this paper. We have changed the Section name from "Description of the detector" to "Observational data" to describe the data set we used.
.
10. It remains unclear from where the lightning data comes. The signals from the Calcutta station are insufficient. It should be explained that – as I assume - the network owner provides processed

data to the authors, i.e. stroke or event listing. Insofar, also the acknowledgement is misleading.

Ans: As we mentioned earlier also, normally we get the processed data i.e., list of all lightning flashes in the region of study (87.65◦ E–89.52◦ E, 22.13◦ N–22.92◦ N) which includes parameters like location, peak current, multiplicity etc. Here in the revised version, we have added, just to get an idea of what the lightning sensor is actually measuring, the raw data of electric field waveform for 1 minute time interval and identified the four types of flashes in that data from our station. We also revised the acknowledgement section.

11. Fig. 2: it is claimed that strokes, grouped to flashes, are shown. Again, the question arises how VHF signals are taken into account. The reader can not understand what the authors have really plotted.

Ans: We have actually plotted the total lightning count per day (Figure 2 of previous version, Figure 3 in the revised manuscript) which includes both IC and CG counts for the month of April, 2018 over the geographic area under investigation.

For VHF signals, we have also mentioned in earlier answers that by extending the frequency range of detection into the MF and HF spectrums, the ENTLN aims to detect and report weaker pulses at longer distances than other VLF/LF systems with similar baselines, since the primary focus of the ENTLN is to maximizing the detection efficiency for cloud flashes.

12. P. 4, line 7: in extension of the previous points, the term "total lightning" needs an explanation. Presumably, it is not just the combination of CG and IC strokes in the VLF/LF range, because VHF signal somehow contribute in a totally unexplained manner. Finally, the question arises, how 1 Hz signals, or ELF signals contribute. In the opinion of this referee, ELF does not matter at all, but it is the authors obligation to communicate in full the used techniques and procedures, and to remove misleading or unnecessary parts.

Ans: Total lightning is the sum of IC and CG flash count as mentioned in the manuscript, identified by the ENTLN using the whole electric waveforms data from various sensors using the methodology described earlier or in the manuscript. We are not using individual strokes in our analysis. There is no ambiguity and we are using the lightning flash data as processed by the ENTLN.
As we mentioned in earlier that, we wish to refer the Earth Network lightning location principle, their manuals and research papers that used ENTLN data. We cannot comment on their sensor technology which uses wide frequency range from ELF to HF range since this is not available or under-presented in the literature.

13. P. 6, line 1: it does not make much sense to add all peak currents of all CG strokes in a storm, because the strokes occur independent of each other at very different locations and the size of the cells may largely vary. It suffices to characterize storm severity by the number of strokes per time and per area.

Ans: We agree with the referee in this point. But still we want to keep this figure in the manuscript to show the difference between amounts of charge transferred involved in the two storms. The numbers of CG flashes are roughly the same for both the storms, but this figures shows the difference between the storms with respect to peak current.

14. P. 6, line 13 and Fig. 5: the multiplicity needs a word on the lower threshold of currents that are determined. The authors should show an additional graph with the distribution of currents for the storm.

Ans: Number of strokes per flash is termed as multiplicity. The ENTLN uses the thresholds of maximum temporal separation of 700 ms and maximum lateral distance of 10 km radius between successive strokes for converting stroke data into flashes.
We do not have the current data for the storms.

15. P. 11, line 1, chapter 5, Conclusion: again, it is not acceptable to claim that installation of one single station allows for monitoring total lightning. In fact, a large number of stations is required and the shown results could also be obtained without the Calcutta station. In total, the paper needs substantial rewriting, because the used instruments, analysis specifications and data handling are not described, preventing the reader from understanding what has been done.

Ans: We have revised the manuscript considering this point and removed the sentences which could lead to misconception or misunderstanding of getting total lightning data from a single station. Keeping this in mind, we have also revised the manuscript title slightly as following:

"A preliminary study on thunderstorms and monsoon using total lightning and weather data over Gangetic West Bengal"

---

## Author Comment (AC4) · 7 Dec 2018

**A preliminary study on thunderstorms and monsoon using total lightning and weather data over Gangetic West Bengal**

Subrata Kumar Midya[1], Sujay Pal[1], Reetambhara Dutta[1], Prabir Kumar Gole[1], Upal Saha[2], Goutami Chattopadhyay[1], Subrata Karmakar[1], and Soumyajit Hazra[3]

[1]Department of Atmospheric Sciences, University of Calcutta, Kolkata-700019
[2]National Centre for Medium Range Weather Forecasting, Noida-201309, India
[3]Skymet Weather Services Pvt. Ltd., Cyberone Tower, Sector-30 A, Vashi, Navi Mumbai

**Correspondence:** Sujay Pal (myselfsujay@gmail.com)

**Abstract. We present a preliminary study of total lightning characteristics of thunderstorms over Gangetic West Bengal around Kolkata. Total lightning data is obtained from the Earth Networks Total Lightning Network (ENTLN) operating in this region in which we are also contributing by hosting a total lightning sensor.** The set up provides improved measurement of **high resolution electric field waveforms corresponding to** in-cloud (IC) as well as cloud-to-ground (CG) lightning discharges in addition to daily weather data **and therefore named as total lightning detector cum-mini weather station (TLDWS).** Severe weather such as thunder squall, Nor'wester, hailstorm, cyclone over the Gangetic West Bengal can be studied in details based on total lightning activity along with other atmospheric and meteorological research using the weather data. We present some analysis of total lightning **data** during the recent Nor'wester events of 2018 **that** occurred in and around Kolkata. **Promising results are obtained as the number of total lightning tends to increase about ∼10–40 minutes before the onset of severe and damaging winds.** We also present variation of **water vapor pressure in air and** wet component of atmospheric refractivity index during the monsoon season which can be used to declare the onset and withdrawal time of monsoon over Gangetic West Bengal.

**1 Introduction**

Thunderstorm and lightning in the troposphere are the most significant atmospheric phenomena which keep life functioning on the earth and at the same time are incredibly destructive to human society in many ways. Lightning discharge radiates electromagnetic energy in a very wide radio frequency range, from below 1 Hz to near 300 MHz, with a maximum radiation energy in the frequency spectrum near 5 to 10 kHz (Rakov and Uman, 2003). Lightning discharge also radiates energy in the optical band $10^{14}$ to $10^{15}$ Hz visible to naked eye which enables ground based camera and satellite to take photograph of lightning event. Further, lightning produce X-rays and *gamma*-rays though not detectable at ground level. Because of this wide range of radiation frequency, there are many ground based and satellite based methods to monitor lightning activity in the atmosphere. Recently, there has been increasing interest in ground based lightning detection networks because of its potential use in meteorological applications. Lightning data from such lightning detection networks are useful for severe weather prediction such as high wind storms, tornadoes, flash flood, hailstorm etc. in addition to study of transient luminous events in the middle

atmosphere. Continuous thunderstorm identification and tracking by lightning detection network also improves the now-casts of thunderstorm, precipitation, severe weather, turbulence and tropical cyclone intensity which can act as a radar proxy in areas of poor radar coverage (Liu and Heckman, 2011; Liu et al., 2014; Heckman et al., 2014). There are several ground based lightning detection networks operating globally for example, Earth Networks Total Lighting Network (ENTLN) which uses wideband electrical field recorders (1 Hz to 12 MHz), Worldwide Lightning Location Network (WWLLN) based on ground based VLF (3-30 kHz) detection of lightning sferics and European VLF/LF lightning detection network LINET.

In a cloud-to-ground discharge, electrical charge is effectively transferred from cloud to ground and normally known as CG lightning. CG lightning can be downward negative, upward negative, downward positive or upward positive discharge. Generally, about 90% or more global CG lightning are downward negative discharge and that of 10% or less of CG lightning are downward positive (Rakov and Uman, 2003). Electrical charge is not transferred from cloud-to-ground in all cases. In fact, majority of lightning discharge, almost 70% or more, occur within the cloud that do not involve ground, known as in-cloud (IC) lightning which can be intra-cloud, inter-cloud and cloud-to-air discharge. In a thunderstorm, central and top parts of the cloud produce IC flashes during the initial stage of electrification, which can be enormously high for a severe storm (Williams et al., 1989). CG flashes increases with more storm electrification during active stage. Strong up-drafts during a storm produce high IC and positive CG flash rates which are the characteristics of a severe thunderstorm (Lang et al., 2000). Recent studies have shown that increase in total lightning i.e., IC and CG flash rate together can produce severe thunderstorm alert which generate high wind, hail storm, tornadoes with a sufficient lead time ranging from 10 minutes to 1 hour (Liu and Heckman, 2011; Liu et al., 2014).

**Thunderstorms and hailstorms in Gangetic West Bengal region were also studied by Doppler weather radar and upper air data with a goal to find thresholds values of different convective indices for thunderstorm prediction (Pradhan et al., 2012). There are also several works on lightning activity over Indian sub-continent due to thunderstorms which have been obtained mainly from satellite borne data such as LIS/TRMM satellite with coarse resolution (Nath et al., 2009; Kumar and Kamra, 2013; Murugavel et al., 2014; Saha et al., 2017; Singh et al., 2014). Few studies on lightning electric field characteristics associated with thunderstorms over North-East India are also prsented using electric field mill data (Guha and De, 2009; Pawar et al., 2010). Midya et al. (2011, 2013b) presented the variation of atmospheric refractive index before, after and during the onset of Nor'westers and squalls over Gangetic West Bengal and shown that sharp depletion of refractive index before the onset of Nor'westers and squall.** In this paper, we have studied total lightning activity during two recent pre-monsoon summer thunderstorms, locally known as "Nor'wester", over the Gangetic West Bengal (GWB) around Kolkata which has not been done before **with respect to total lightning analysis**. Nor'wester is the short duration severe thunderstorms with high wind speed occurring every year during late March to May in the eastern and north-eastern part of India including Bangladesh. This brings considerable damage to agriculture, properties and even human life. Large number of lightning associated with the Nor'wester during the active stage of thunderstorm are the main reason of fatalities in this region. This paper attempts to study the total lightning activity during Nor'wester events from formation stage to dissipation stage with emphasis on short-term prediction of severity of the storm.

We have also studied the variation of wet component of refractivity index **and water vapor pressure** during monsoon period using the data from the total lightning detector-cum-mini weather station (TLDWS) to find possible onset and withdrawal signature for monsoon over Kolkata. The monsoon specially south west monsoon is an important atmospheric circulation which affects the life and economy of Indian subcontinent. Traditionally monsoon is defined as the seasonal reversal of wind pattern associated with heavy precipitation. June, July, August and September are the principal monsoon months over Indian subcontinent. Indian summer monsoon rainfall (ISMR) is the rainfall carried by the south-west monsoon during June to September every year and accounts for approximately 80% of the annual rainfall over India. In recent years, people attach importance to study of monsoon rainfall variations and proper prediction of onset and withdrawal of monsoon. Various studies represented the ISMR change is related to some meteorological parameters like surface temperature (Chattopadhyay et al., 1995), relative humidity, sea level barometric pressure (Parthasarathy et al., 1992; Bansod et al., 1995). El Nino Southern Oscillation (ENSO) events (Mooley et al., 1985; Gadgil et al., 2004), Sea Surface Temperature (Nicholls, 1995; Sahai et al., 2003; Rai and Pandey, 2008), Quasi Biennial Oscillation (QBO), Cloud condensation nuclei counter, aerosol concentration and even relative sunspot number (SSN) and Flare index (Hiremath and Mandi, 2004) have significant impacts on monsoon. The accumulated impacts of these various parameters make monsoon prediction more complicated and more challenging task. As monsoon is the principle rainy season over Indian sub-continent, proper prediction of onset and withdrawal of monsoon is very crucial. Climatologically monsoon onset takes place over Kerala (a Southern state in India) on 1st June and over Kolkata on 10th June. By the end of June, it covers more than 90% of the area of India and by mid-July the whole of India is covered by the monsoon. In early September, summer monsoon rains begin to withdraw from north-west part of India and from entire country by mid-October. During monsoon onset, dramatic changes of large scale atmospheric structure are known to occur over India. Some of the well known ones associated with onset are rapid increase in daily rain rate, increase in the vertically integrated moisture and the increase in the strength of low level monsoon flow. Many researchers have studied onset and withdrawal of monsoon in India using various parameters, such as outgoing long wave radiation (OLR), integrated water vapor (IWP), low level jet stream (LLJ), sea surface temperature (SST) (Joseph et al., 1994, 2006), wind data (Wang et al., 2009) and vertically integrated moisture transport variability (Fasullo and Webster, 2003). Using GPS radio occultation data for 2001-2010, Rao et al. (2013) examined variation of atmospheric refractivity during the onset of ISM over east Arabian sea and observed an enhancement of 5-10 N units in refractivity a few days before on set of monsoon over Kerala. Till today, scientists are trying to find out more and more reliable parameters for prediction of exact onset and withdrawal of monsoon. Midya et al. (2013a, b) reported that the variation of wet component of refractivity gives an indication of cyclonic movement and onset of squall over Kolkata. In addition to study the total lightning activity during Nor'wester days, we have also examined the variation of wet component of refractivity **and water vapor pressure**, in this paper, during monsoon season to check possible signature of monsoon onset and withdrawal time.

[Figure]

**Figure 1.** Earth Networks total lightning detector-cum-weather instrument: (a) Wind speed and direction sensor (b) Integrated in-cloud (IC) and cloud-to-ground (CG) lightning detection sensor (c) Sensor Shelter (d) Rain Gauge (e) Lightning Remote Box (f) Network appliance. **Lightning sensor is a part of ENTLN**.

**2   Observational Data**

Earth Networks total lightning detector-cum-mini weather station (TLDWS), **has been operating in our Kolkata station since June 2016 for monitoring of various weather parameters and recording of lightning electric field data. Electric field data are being used in the ENTLN for lightning location purpose.** The TLDWS consists of several parts (shown in Figure 1) listed below: (a) Weather sensor which captures wind speed and direction; (b) Integrated in-cloud (IC) and cloud-to-ground (CG) lightning detection sensor which mainly measures electromagnetic signals from lightning discharge; (c) Sensor Shelter which measures mainly temperature, relative humidity, heat index, wind chill, barometric pressure and dew point; (d) Rain gauge which measures daily, monthly and yearly rainfall totals and averages; (e) Lightning remote box (f) Network appliance which is basically an IP-enabled device that connects easily to the internet, provides fast transmission of the data to the server. It has a 72-hour battery life and automatically reboots as needed.

The integrated IC and CG lightning sensor operates in a frequency range from 1 Hz to 12 MHz (spanning the ELF, VLF, LF, MF, and HF ranges) and measures the electromagnetic signals from each lightning discharge (Heckman et al., 2014). **The**

primary focus of the ENTLN appears to maximize the detection efficiency for cloud flashes. The ENTLN claims to detect weaker pulses at longer distances than other VLF/LF systems with similar baselines by extending the frequency range of detection into the MF and HF spectrums (Heckman and Liu, 2010 ). The whole electric field waveforms are transferred from the **sensor** to the central data processor of Earth Networks via internet and network appliance. Central processor will then geolocate the individual lightning event and calculate the associated lightning parameters (such as peak current, multiplicity, lightning types etc.) from the waveform characteristics **sent** by this sensor and other sensors in the **network**. Time of arrival (TOA) method is being used to geolocate lightning event. In this method the onset time, arrival time, time of peak magnitude of a lightning pulse measured by multiple sensors (at least four) are analyzed at the central processor to determine the four unknowns latitude, longitude, height and time that define the source location (Heckman, 2014). Each lightning discharge consists of several strokes. In the ENTLN, individual strokes occurring within 700 ms and 10 km of the first stroke detected by the sensors are clustered into a flash. A flash is further classified as a CG flash if it contains at least a return stroke, otherwise it is classified as a IC flash. **Typical recording of electric field amplitude is presented in Figure 2, to get an idea what the lightning sensor is measuring. Here the top panel shows the raw measurements of electric field amplitude data for 1 minute time interval after 14:15 UT of 17th April, 2018. Electric field amplitude is in raw digitizer units. Other four panels show the variation of electric field waveform corresponding to four types of lightning discharge as identified by ENTLN. Distances from the lightning location to the receiving station corresponding to the four discharges are 50.50 km (for +CG), 43.80 km (for -CG), 43.89 km (for +IC) and 7.80 km ( for -IC) respectively. Note the presence of stepped leader (SL) and return stroke (RS) in case of CG lightning discharge.** In Figure 3, we have presented the total lightning count (IC+CG) per day for April, 2018 **over the study area bounded by** 87.65°E–89.52°E and 22.13°N–22.92°N. High lightning count greater than 10,000 are the Nor'wester days over this region.

**3   Nor'wester and total lightning activity**

Here we report the preliminary results **of analysis of total lightning activity during pre-monsoon summer thunderstorms over the study area corresponding to two Nor'wester events that occurred on April 7 (event 1) and April 17, 2018 (event 2).** Event 1 was a non-severe type having maximum wind speed 64 km/hr with no causalities reported. Whereas around 15 people from Kolkata and adjacent districts were **killed** (Source: The Hindu) during the Nor'wester of 17th April with maximum wind speed of 98 km/hr recorded by IMD (Indian Meteorological Department), Kolkata between 19:42 IST to 20:15 IST.

A total of 13,242 lightning flashes during 17:00 IST to 23:00 IST on April 7, of which 9,243 were in-cloud (IC) lightning (69.80 %) and 3,999 were cloud-to-ground (CG) lightning (30.20%) **were recorded by the ENTLN**. On April 17, **ENTLN recorded** a total of 10,541 flashes during 17:30 IST to 22:00 IST, of which 7,403 were in-cloud (IC) lightning (70.24 %) and 3,138 were cloud-to-ground (CG) lightning (29.76%). Out of all CG flashes, almost 12% flashes were positive CG for the event 1, while 17% flashes were positive CG for the Nor'wester event 2 as classified by the **ENTLN. Figure 4a (top panel, left) and 4c (lower panel, left) show the locations of all the lightning flashes (IC, red dots and CG, green dots) over the geographical area under this study for the two events. Figure 4b (top panel, right) and 4d (lower panel, right) show the**

[Figure]

**Figure 2. Top panel shows the raw measurements of electric field amplitude data for 1 minute time interval after 14:15 UT of 17th April, 2018. Electric field amplitude here is in raw digitizer units. Other four panels show the variation of electric field waveform corresponding to four types of lightning discharge as identified by ENTLN.**

**evolution of different types of lightning flash rate (number of flashes per minute) as detected by the ENTLN within the area** for the event 1 and event 2 respectively. Black lines show CG flash rate, red line show IC flash rate and blue lines represent total lightning (CG+IC) flash rate per minute. Thick lines are the 7 point running mean curves. For both events, total lightning flash rate increased drastically to about 110-120 flashes per minute during the active stage, from about 20-30 flashes per minute

5    during the initial stage of thunderstorms showing lightning jump. Lightning jump in the flash rate are used to predict the severe thunderstorms well ahead of the peak damaging wind and hailstorm [Williams et al. 1999]. In Figure 4, dashed vertical lines are used to identify the time when Nor'wester hit the region with peak wind speed recorded by IMD, Kolkata. Therefore, it is obvious that study of total lightning activity can be a good indicator for the dangerous Nor'wester events well ahead of time to mitigate the damages caused by them.

10    The thunderstorm of 17th April ended with a sudden decrease in IC flash rate but the thunderstorm of 7th April shows gradual decay of both IC and CG flash rates. It is also to be noted that the CG flash rate peaked before the IC flash rate and damaging wind for the 2nd event, but the IC and CG flash rates simultaneously increased and decreased during the storm lifetime for the 1st event. Now to identify the severity of the thunderstorm using lightning characteristics, we have calculated total discharge peak current per minute by the positive and negative CG flashes, since the CG flashes cause most damage to

15    human life, power lines and consumer electronics on earth. Figure 5 shows the temporal distribution of total peak current per

[Figure]

**Figure 3.** Total lightning flash count (IC+CG) per day **for the month of April, 2018 in the region of study bounded by (87.65°E–89.52°E, 22.13°N–22.92°N).**

minute (kA/min) due to positive and negative CG flashes for the two events respectively. Solid lines are the 7 point running mean curves. While it is difficult to identify the intensity of the storm from the total peak current per minute from the negative CG flashes, we can see that mean discharge current was more than 100 kA/min touching to 200 kA/min by the positive CG flashes during the active phase of thunderstorm on 17th April and the same for the thunderstorm of 7th April was merely 100
5    kA/min. Also for the 2nd event, 5% more positive CG flashes occurred than the 1st event. Therefore, positive CG lightning can be used to identify severity of thunderstorm event. We are analysing all the thunderstorm events over GWB based on total lightning activity and results will be reported in due time.

     Another important characteristic of lightning is the stroke multiplicity which cause damage to human life and consumer electronics along with peak current **(Miyazaki and Okabe, 2008; Gibbs, 2012)**. A lightning flash normally **consists** of one or
10    several strokes which are within 700 ms and 10 km of the first stroke as detected by **ENTLN**. The number of strokes in a flash is known as lightning multiplicity. Figure 6 shows the multiplicity distribution of negative and positive CG flashes occurred on April 7 and April 17, 2018 during the Nor'wester events. We note that 59% of negative CG flashes are composed of a single stroke for the thunderstorm of April 7, 2018 whereas 72% of negative CG flashes are composed of a single stroke for the severe thunderstorm of April 17, 2018. Average multiplicity of negative CG flashes are found to be 1.96 for April 7 and 1.64 for April
15    17, 2018, while the same for positive CG flashes are found to be 1.09 for April 7 and 1.22 for April 17, 2018 respectively.

     It is also possible to track the life cycle of a thunderstorm cell using the total lightning activity which is frequently done by the weather radar system. When the lightning flash rates are high enough, thunderstorm cells can be identified with total flashes

[Figure]

**Figure 4. Locations of all the IC (red dots) and CG (green dots) flashes over the study region and corresponding temporal evolution of lightning flash rate (per minute) during the two recent Nor'wester events. Blue circle indicates the location of our station.**

occurring in clusters which can be be used for early warning of severe storms [Betz et al. 2008; Liu et al. 2014]. In Figure 7, we present an example of time evolution of lightning cells as snapshots of total lightning activity for 10 minutes time interval in the GWB region around Kolkata for both the thunderstorm events. Note that the lightning cells come from the north-westerly direction which gives its name as Nor'wester.

5 **4 Atmospheric refractivity and monsoon over Kolkata**

Here we present the variation of wet component of atmospheric refractivity index ($\eta_w$ in ppm) **and water vapor pressure** during monsoon period over Kolkata during 2016 using the data from TLDWS to show the **potential of these two parameters**

[Figure]

**Figure 5.** Temporal distribution of total peak current (kA) per minute due to ±CG lightning during the thunderstorms of 7th April and 17th April.

[Figure]

**Figure 6.** Flash multiplicity for positive and negative CG lightning flashes.

[Figure]

[Figure]

**Figure 7.** Temporal evolution of thunderstorm cells during the two Nor'wester events **of 7th April, 2018 (left) and 17th April, 2018 (right) over the GWB**. Note that the thunderstorm cell comes from the north-westerly direction.

**as the tools** to declare the onset and withdrawal dates of monsoon. Variation of wet component of atmospheric refractivity with time has been studied using the formula for troposphere as used by Midya et al. (2013a). Mainly atmospheric pressure, temperature and relative humidity are used to calculate ($\eta_w$). We have taken the wet component of refractivity because it is highly dependent on the presence of water vapor in the atmosphere. As the dry component of refractivity is not dependent on
5   the presence of water vapor in the atmosphere, we have neglected this term. The variation of **wet component of refractivity (first panel), water vapor pressure (second panel), temperature (3rd panel, red) and relative humidity (3rd panel, blue) on hourly basis from June to December of 2016 are plotted in Figure 8. Straight line joining day number 176 to day number 201 (June 24 to July 19, 2016) represents data gap when TLDWS was not functioning.** The figure shows a steady higher value of **refractivity and water vapor pressure during monsoon and a sharp decrease in both quantity during**
10   **withdrawal of monsoon over Kolkata. Up arrow in the first panel of Figure 8 shows the date of monsoon withdrawal over the Gangetic West Bengal as declared by the IMD.** It can be also clearly seen that during the monsoon period the daily fluctuations in refractivity and vapor pressure also reduced. Variation of wet component of refractivity from two data sources, the TLDWS and IMD, Kolkata (VECC station), are compared in Figure 9 and exactly the same variation is seen.

Monsoon is the reversal of wind pattern associated with heavy precipitation. In India two types of monsoon can be seen,
15   one is summer monsoon or south-west monsoon and another is winter monsoon or north-east monsoon. Gangetic West Bengal receives south-west monsoon dominantly, but north-east monsoon can hardly be observed. In north hemispheric summer, due to differential heating of landmass and ocean body, a low pressure develops over interior of Asia as well as over North-Western India. At the same time a high pressure region persists over the Southern Indian Ocean. As a result winds blows from this high pressure region to low pressure region. After crossing the equator, due to Coriolis force, this wind turns into right and starts
20   flowing from the South-West direction and enters into Indian Peninsula. During the journey of this wind over warm Tropical Ocean, it acquires abundant moisture within it. When it arrives near the southern tip of the Indian peninsula, this wind system breaks into two branches. One is the Arabian Sea branch which hits the Western Ghats, and another is the Bay of Bengal branch

[Figure]

**Figure 8. Variation of wet component of atmospheric refractivity index (1st panel), vapor pressure of water (2nd panel), temperature (3rd panel, red) and relative humidity (3rd panel, blue) during June to December 2016 obtained from the TLDWS are shown here. Straight line joining day number 176 to day number 201 (June 24 to July 19, 2016) represents data gap when TLDWS was not functioning. Up arrow in the 1st panel shows the date of monsoon withdrawal over the GWB as declared by the IMD.**

[Figure]

**Figure 9. Comparison of wet component of atmospheric refractivity index obtained from IMD (VECC) station (black) and TLDWS (red). Data are not available from the TLDWS before June, 2016**.

which flows over the Bay of Bengal and hits the eastern Himalaya. Thus during summer this surface westerly wind (blowing from south west direction) brings ample amount of water vapor form the Bay of Bengal into the GWB basin. As Partial vapor pressure depends on relative humidity, wet component of refractivity noticeably increases and indicates the increased water vapor content in the atmosphere, when the moist air from Bay of Bengal enters. When this water vapor condenses, heavy precipitation occurs in this region. As water vapor is the primary source of precipitation, the onset of monsoon is expected to occur over GWB when sufficient amount of water vapor has been carried from the Bay of Bengal by the westerly wind into the GWB basin. Similarly when surface easterly blows dominantly, the amount of water vapor reduces over the GWB basin and monsoon is expected to withdraw. From Figure 7 it is seen that a sharp decrease in refractivity occurred on 13.10.16, so it may be monsoon withdrawal date in this year. IMD declares 16.10.2016 as the withdrawal date in GWB. IMD declares the dates on the basis of some criteria given later and in our study only wet component of refractivity is considered.

IMD which is the principal Government Department of Weather Forecasting declared onset and its further advancement over the country with three criteria given below. (a) Rainfall: If after 10th May, 60% of the available 14 stations viz. Minicoy, Amini, Thiruvananthapuram, Punalur, Kollam, Allapuzha, Kottayam, Kochi, Thrissur, Thalassery, Kannur, Kudulu and Mangalore report rainfall of 2.5 mm or more for two consecutive days, the onset over Kerala can be declared on the 2nd day provided the following criteria are also in concurrence. (b) Wind Field: Depth of westerlies should be maintained upto 600 hPa, in the box equator to Latitude 10°N and Longitude 55°E to 80°E.The zonal wind speed over the area bounded by Latitude 5-10°N, Longitude 70-80°E should be of the order of 15-20 Kts. (c) Long wave radiation (OLR): INSAT derived OLR value should be below 200 $wm^{-2}$ in the box confined by Latitude 5-10°N and Longitude 70-75°E.

Withdrawal of monsoon are declared on the basis of reduction in moisture and prevalence of dry weather for 5 days. It is quite expected that the refractivity increases with increase of water vapor content in the atmosphere. During monsoon period, when the water vapor enters over Gangetic West Bengal, refractivity increases significantly. Similarly when the water vapor is withdrawn, monsoon disappears. But the fact is that, presence of water vapor is not only the criterion of onset of monsoon. Presence of cloud condensation nuclei (CCN), dew point temperature etc. are also essential criteria to start rainfall **(Midya et al., 2015; Ganda and Midya., 2012)**. In our observation refractivity becomes maximum when water vapor enters over GWB and during monsoon period it remains at a higher steady value because during whole monsoon period there is an ample amount of water vapor supply. And during withdrawal of monsoon it sharply decreases from the higher steady value. Therefore, study of wet component of atmospheric refractivity can be used as a tool to declare the onset and withdrawal dates of monsoon.

**5 Summary and Conclusion**

**In this paper we have studied pre-monsoon thunderstorm events over Gangetic West Bengal using the total lightning data and monsoon characteristics with respect to water vapor pressure.** We have shown the usefulness of **total lightning data** to predict the high and damaging wind corresponding to Nor'wester events around Kolkata. In our initial study, we have found that total lightning flash rate which includes both IC and CG flash rate starts increasing rapidly during the initial stage of

the thunderstorm much before the high wind and high peak current CG lightning occurred. The severity of Nor'wester storm can also be predicted from the characteristics of IC lightning and positive CG lightning. More works are needed to establish the relationship of total lightning characteristics with damaging wind, dangerous lightning, heavy rainfall and hailstorm associated with Nor'wester events in this region. We have also shown that onset and withdrawal of Indian monsoon over **the Gangetic**

5 **West Bengal region can be studied from the variation of water vapor pressure and atmospheric refractivity index**. In summary, the total lightning provides a very good opportunity to study severe weather associated with thunder squall, Nor'wester, hailstorm, cyclone, heavy precipitation as well as to study various other meteorological and atmospheric research in Gangetic West Bengal.

*Acknowledgements.* We thank the Earth Networks for providing the ENTLN data and for supporting the TLDWS at our location. S. Pal

10 would like to thank Dr. Michael Stock of Earth Networks for a fruitful discussion about ENTLN. S. Pal also acknowledges the support from the University Grants Commission (UGC) under the Dr. D.S. Kothari Fellowship Scheme (No.F.4-2/2006(BSR)/ES/17-18/0007).

**References**

Bansod, S. D., and Singh, S. V.: Pre-monsoon surface pressure and summer monsoon rainfall over India, Theor. Appl. Climatol., 51, 1995.

Betz, H. D., Schmidt, K., Oettinger, W. P., and Montag, B.: Cell-tracking with lightning data from LINET, Adv. Geosci., 17: 55–61, 2008.

Chattopadhyay, J., Pandey, S. N., and Banerjee, M.: The coherence between surface air temperature and Indian monsoon rainfall, Pure and
    Applied Geophysics, 144-1, 155–165, 1995.

Fasullo, J. and Webster, P. J.: A hydrological definition of Indian monsoon onset and withdrawal, J. of Climate, 16, 3200-3211, 2003.

Gadgil, S., Vinayachandran, P. N., Francis, P. A., and Gadgil, S.: Extremes of the Indian summer monsoon rainfall, ENSO and equatorial
    Indian Ocean oscillation, Geophy. Res. Let., 31(12), L12213, 2004.

Ganda, S.C., and Midya, S.K.: Comparison of long term rainfall trends on urban and non-urban regions of Indian land mass and its probable
    impliction, J. Ind. Geo. Union, 16(2), 37–40, 2012.

Gibbs, Alexander R.: Periodicities of Peak Current and Flash Multiplicity in Cloud to Ground Lightning, Dissertations & Theses in Earth
    and Atmospheric Sciences, 24, 2012, http://digitalcommons.unl.edu/geoscidiss/24

Guha, A., and De, B.K.: A Preliminary Study of Intra-Cloud Lightning Electrical Characteristics during Tropical Summer Thunderstorm in
    North-East India, URSI GASS Proceedings, 2008.

Heckman, S., and Liu, C.: The application of total lightning detection and cell tracking for severe weather prediction, in Proc. of
    GROUND'2010 & 4th LPE, Salvador, Brazil, 234–240, 2010.

Heckman, S.: ENTLN status update, XV International Conference on Atmospheric Electricity, Norman, Oklahoma, U.S.A., 15-20 June 2014.

Heckman, S., Liu, C., and Sloop, C.: Earth networks lightning overview, International Conference on Lightning Protection (ICLP), IEEE,
    10.1109/ICLP.2014.6973433, 2014.

Hiremath, K. M., and Mandi, P. I.: Influence of the solar activity on the Indian Monsoon rainfall, New Astronomy, 9(8), 651–662, 2004.

Joseph, P. V., Eischeid, J. K., and Pyle, R. J.: Inter-annual variability of the onset of Indian summer monsoon and its association with
    atmospheric features, El Nino and sea surface anomalies, J. of Cimate., 7, 81-105, 1994.

Joseph, P. V., Sooraj, K. P., and Rajan, C. K.: The summer monsoon onset process over South Asia and an objective method for the date of
    monsoon onset over Kerala, Int. J. of Climatology, 26, 1871-1893, 2006.

Kumar, P.R., and Kamra, A.K.: The lightning activity associated with the dry and moist convections in the Himalayan regions. J. Geophys.
    Res. Atmos. 118, 6246–6258. http://dx.doi.org/10.1002/jgrd.50499, 2013.

Lang, T. J., and Rutledge, S.A., Dye, J. E., Venticinque, M., Laroche, P., and Defer, E.: Anomalously low negative cloud-to-ground lightning
    flash rates in intense convective storms observed during STERAO-A, Mon. Wea. Rev., 128, 160–173, 2000.

Liu, C., and Heckman, S.: The Application of Total Lightning Detection and Cell Tracking for Severe Weather Prediction, Session presented
    at the 91st American Meteorological Society Annual Meeting, Seattle, WA, 2011.

Liu, C., Sloop, C., and Heckman, S.: Application of lightning in prediction high impact weather, OBS/IMO/TECO-2014.

Midya, S. K., Sarkar, H. and Saha, U.: Sharp depletion of atmospheric refractive index associated with Nor'wester over Gangetic West
    Bengal: a possible method of forecasting Nor'wester, Meteorol. Atmos. Phys. 111 149–152, 2011.

Midya, S. K., Das, G. K., and Sarkar, A.: The relationship between wet component of atmospheric refractivity and movement and landfall of
    tropical cyclone in the Bay of Bengal region, Meteorol. Atmos. Phys., 121, 153–159, 2013a.

Midya, S. K., Ghosh, D., Das, G. K., and Sarkar, H.:Study of atmospheric refractivity prior to squall onset and its strong association with
    surface temperature and relative humidity over Kolkata, Indian J. Phys., 87(9), 847–854, 2013b.

Midya S. K., Ghosh S., Ganda S.C., and Das, G.K.: Role of Biogenic Hydrocarbon on the Variability of Total Rainfall Amount over Sundar-ban, Kaziranga and Gir Forests, J. Ind. Geo. Union, 19, 454–459, 2015.

Miyazaki, T., and Okabe, S.: A Detailed Field Study of Lightning Stroke Effects on Distrubtion, IEEE Transactions on Power Delivery, 24, 352–359, 2008.

5  Mooley, D. A., Parthasarathy, B., and Sontakke, N. A.: Relationship between all-India summer monsoon rainfall and southern oscillaiton/east equatorial pacific sea surface temperature, Proc. Indian Acad. Sci. (Earth Planet. Sci.), 94, 199–210, 1985.

Murugavel, P., Pawar, S.D. and Gopalakrishan, V.: Climatology of lightning over Indian region and its relationship with convective available potential energy, Int. J. Climatol. 34: 3179–3187, 2014.

Nath A., Manohar, G.K., Dani, K.K. and Devara, P.C.S.: A study of lightning activity over land and oceanic regions of India, J. Earth Syst.
10  Sci. 118, No. 5, 467–481, 2009.

Nicholls, N.: All-India summer monsoon rainfall and sea surface temperatures around Northern Australia and Indonesia, J. Climate, 8, 1463–1467, 1995.

Parthasarathy, B., Kumar, K. R., and Munot, A. A.: Surface Pressure and Summer Monsoon Rainfall over India, Adv. Atmos. Sci., 9(3), 359–366, 1992.

15  Pawar, S.D., Murugavel, P., and Gopalakrishnan, V.: Anomalous electric field changes and high flash rate beneath a thunderstorm in northeast India, J. Earth Syst. Sci. 119, 5, 617–625, 2010.

Pradhan, D., De, U.K., Singh, U.V.: Development of now-casting technique and evaluation of convective indices for thunderstorm prediction in Gangetic West Bengal (India) using Doppler Weather Radar and upper air data, MAUSAM, 63, 2, 299-318, 2012.

Rai, S., and Pandey, A. C.: Southern Indian Ocean SST variability and its relationship with Indian summer monsoon, Atmosphere-Ocean,
20  46(3), 361–376, 2008.

Rakov V. A., and Uman M. A.: Lightning: Physics and effects, Cambridge University Press, Cambridge, U. K., 2003.

Rao, V.V.M.J., Ratnam, M. V., Santhi, Y. D., Raman, M. R., Jajeevan, M., and Rao, S.V.B.: On the detection of onset and activity of the Indian Summer Monsoon using GPS RO Refractivity profiles, Am. Meteor. Soc., 141, 2096-2106, 2013.

Saha, U, D. Singh, S.K. Midya, R.P. Singha, A.K. Singha, S. Kumar: Spatio-temporal variability of lightning and convective activity over
25  South/South-East Asia with an emphasis during El Niño and La Niña, Atmospheric Research, 197, 150-156, 2017.

Sahai, A. K., Grim, A. M., Satyan, V., and Pant, G. B.: Long-lead prediction of Indian summer monsoon rainfall from global SST evolution, Climate Dynamics, 20, 855-863, 2003.

Singh, D., Buchunde, P.S., Singh, R.P., Nath, A., Kumar, S., Ghodpage, R.N.: Lightning and convective rain study in different parts of India, Atmos. Res. 137, 35–48, 2014.

30  Wang, B., Ding, Q., and Joseph, P. V.: Objective definition of the Indian summer monsoon onset, J. of Climate, 22, 3303-3316, 2009.

Williams E.R., Weber, M., and Orville, R.: The relationship between lightning type and convective state of thunderclouds, J. Geophys. Res., 94, 13213–13220, 1989.

Williams, E.R., Boldi, B., Matlin, A., Weber, M. Hodanish, S., Sharp, D., Goodman, S., Raghavan, R., and Buechler, D.: The behavior of total lightning activity in severe Florida thunderstorms, Atmos. Res., 51, 245–65, 1999.